# Lithographic Mask Defects Analysis on an MMI 3 dB Splitter †

**Paulo Lourenço** [1,2,*], **Alessandro Fantoni** [2,3], **João Costa** [2,3] and **Manuela Vieira** [2,3]

1. Departamento de Engenharia Eletrotécnica, FCT, Universidade Nova de Lisboa, Campus da Caparica, Faculdade de Ciências e Tecnologia, 2829-516 Caparica, Portugal
2. CTS-UNINOVA, Departamento de Engenharia Eletrotécnica, Faculdade de Ciências e Tecnologia, FCT, Universidade Nova de Lisboa, Campus da Caparica, 2829-516 Caparica, Portugal; afantoni@deetc.isel.ipl.pt (A.F.); jcosta@deetc.isel.ipl.pt (J.C.); mv@isel.ipl.pt (M.V.)
3. ISEL—Instituto Superior de Engenharia de Lisboa, Instituto Politécnico de Lisboa, Rua Conselheiro Emídio Navarro, 1, 1959-007 Lisboa, Portugal
* Correspondence: pj.lourenco@campus.fct.unl.pt
† This paper is an extended version of our paper published in: SPIE Paper Number: AOP100-40. Lourenço, P.; Fantoni, A.; Vieira, M. Simulation analysis of a thin film semiconductor MMI 3 dB splitter operating in the visible range. In Proceedings of AOP 2019, IV International Conference on Applications of Optics and Photonics, Lisbon, Portugal, 31 May–4 June 2019.

**Abstract:** In this paper, we present a simulation study that intends to characterize the influence of defects introduced by manufacturing processes on the geometry of a semiconductor structure suitable to be used as a multimode interference (MMI) 3 dB power splitter. Consequently, these defects will represent refractive index fluctuations which, on their turn, will drastically affect the propagation conditions within the structure. Our simulations were conducted on a software platform that implements the Beam Propagation numerical method. This work supports the development of a biomedical plasmonic sensor, which is based on the coupling between propagating modes in a dielectric waveguide and the surface plasmon mode that is generated on an overlaid metallic thin film, and where the output readout is achieved through an a-Si:H photodiode. By using a multimode interference 1 × 2 power splitter, this sensor device can utilize the non-sensing arm as a reference one, greatly facilitating its calibration and enhancing its performance. As the spectral sensitivity of amorphous silicon is restricted to the visible range, this sensing device should be operating on a wavelength not higher than 700 nm; thus, a-SiNx has been the material hereby proposed for both waveguides and MMI power splitter.

**Keywords:** a-SiNx; beam propagation method; multimode interference; 3 dB splitter

## 1. Introduction

a-SiNx is considered a promising platform for photonic integrated circuits (PIC) as PICs can be fabricated in state-of-the-art foundries with integrated CMOS electronics [1,2]. Besides, the high index contrast between a-SiNx and silicon dioxide allows the production of submicron cross section waveguides and very small bending radii, hence enabling high density integration. While much of the existing work on CMOS photonics [1,2] has used directional couplers for power splitting, multimode interference (MMI) devices may have relaxed fabrication requirements and smaller footprints [3], when compared to other configurations based on parallel waveguides coupling. MMI devices have already been used as 1 × 2 splitters, 2 × 1 combiners in Quadrature Phase Shift Keying (QPSK) modulators [4], 3 dB couplers in ring resonators [5], and cross couplers for switches [6,7]. Also, state-of-the-art CMOS manufacturing processes may achieve nanometer scale lithographic precision. This work is a sequel

of previous research [8], where MMI tolerances to manufacturing processes are analyzed through simulations and the robustness of the MMI devices is evaluated.

The remainder of this paper is organized as follows:

- Next follows the Materials and Methods section, where will be described a $1 \times 2$ multimode interference (MMI) structure consisting of a 3 dB coupler that provides the required optical paths to feed the sensing device and to yield the reference arm signal for further analysis and processing.
- Then, we present the Results section. In this section, we evaluate the imbalance between the outputs of the MMI structure considering an increasing standard deviation of a normal distribution of defects. These perturbations are intended to represent the defects introduced by lithographic mask resolution along the whole structure. Next, these defects' distribution impacts are also evaluated from a manufacturing perspective where a batch of produced samples with defects is emulated by simulation and the results obtained through statistical analysis are reported.
- Finally, there is the Discussion section where obtained results are interpreted and conclusions are reported. Here, future areas of related research will also be discussed, namely future research/simulation actions that will contribute to mitigating the imbalance introduced by lithographic mask resolution.

## 2. Materials and Methods

This work precedes and provides support for a broader project consisting of the development of a biomedical plasmonic sensor device. This sensing structure relies on the surface plasmon resonance phenomenon, which consists of the development of surface waves at the separation interface between a thin metallic film and a dielectric, to detect minute refractive index variations on the surrounding environment. These surface waves result from the coupling between propagating modes in a dielectric waveguide and the top deposited metal film.

In this biomedical-sensing device project, the detection outcome readout will be achieved through an a-Si:H photodetector to enable a high level of integration and compliance with the lab-on-a-chip concept [9]. As the spectral sensitivity of amorphous silicon is within the visible range, the operating wavelength must not exceed 700 nm. Hence, a-SiNx is the material used throughout the rest of this paper for both input and output waveguides and MMI structure, for it presents a transparent behavior at these wavelength ranges.

Our intent is to obtain two identical electromagnetic (EM) field profiles resulting from an initial fundamental mode input profile. One of the EM fields will be affected by a sensing area and the other one will represent a comparison reference for further processing. This detection area will consist of a thin metallic film deposited atop one of the output waveguides. At the interface between the superstrate and the metallic film, a surface wave will be generated due to the coupling of the propagating modes in the dielectric waveguide and the surface plasmon resonance mode developed on the deposited metallic film. However, for this coupling to occur, it is required that the propagating mode inside the dielectric waveguide possesses the $y$-axis magnetic component ($H_y$) of the EM field, thus only transverse magnetic (TM) modes fulfil this requirement [10]. For this reason, all subsequent discussion and analysis will be considering a propagating TM mode.

The available options evaluated were single mode Y-junction, two-mode interference (TMI) and MMI devices, but former devices presented disadvantages when compared to the latter structure. Namely, close proximity of access waveguides on these devices usually leads to unwanted modal coupling and separation aperture filling, due to the limited resolution of lithographic process. This affects TMI and coupling sections (for TMI and Y-junction devices, respectively) length arbitrarily, thus causing performance degradation [9].

According to literature [11,12], the operation of MMI devices relies on the self-imaging principle which states that single or multiple images of a given input field profile are replicated periodically in space as the electromagnetic field propagates through the waveguide. The propagation constant of a mode $\beta_m$ ($m = 0, 1, 2, \ldots$) propagating in a high contrast step index multimode device shows an approximate quadratic dependence to the mode number $m$:

$$\beta_m \cong k_0 n_{core} - \frac{(m+1)^2 \pi \lambda_0}{4 n_{eff} W_{eff}^2},\tag{1}$$

where $k_0$ is the vacuum wavenumber, $n_{eff}$ the effective refractive index of the structure, $\lambda_0$ the vacuum wavelength, and $W_{eff}$ the effective width of the MMI waveguide. The effective refractive index, $n_{eff}$, is a characteristic of each propagating mode, representing the "experienced" refractive index, when propagating inside the waveguide structure and the standing-wave condition is matched [12]. Hence, the $n_{eff}$ of a given propagating mode may be calculated as in Equation (2) below:

$$n_{eff} = n_{core} sin\theta_p,\tag{2}$$

where $n_{core}$ is the core media refractive index, and $\theta_p$ is the refraction angle of the transmitted field with respect to the normal at the input interface and at the point of reflection. The effective width, $W_{eff}$, is the width when considering the mode field profile penetration depth, due to the Goos-Hähnchen shifts, into the waveguide boundaries. This dimension is polarization dependent and in high refractive index contrast devices, the penetration depth of the EM field beyond the inner walls of the device is practically non-existent, hence $W_{eff}$ can be approximated by the effective width of the fundamental mode [11]:

$$W_{eff} \cong W_{m0}.\tag{3}$$

The spatial location of single/multiple and direct/mirrored images, resulting from the propagation modes interference, is directly related to the beat length ($L_\pi$) of the two lowest order modes:

$$L_\pi = \frac{\pi}{\beta_0 - \beta_1} \cong \frac{4 n_{eff} W_{eff}^2}{3\lambda_0} \cong \frac{4 n_{eff} W_{mo}^2}{3\lambda_0}\tag{4}$$

Single mirrored and direct images from the input field profile form at $3L_\pi$ and $2(3L_\pi)$, respectively, while two-fold images form at $1/2(3L_\pi)$ and $3/2(3L_\pi)$. Single images are, approximately, the same amplitude as the input EM field and each of the two-fold images is affected by a 3 dB attenuation factor, thus offering the ideal conditions for a power splitter device, similar to the structure diagram depicted in Figure 1. This schematic represents an a-SiNx MMI device embedded in SiO$_2$ and associated dimensions, which were used throughout this paper in our simulations and subsequent analysis.

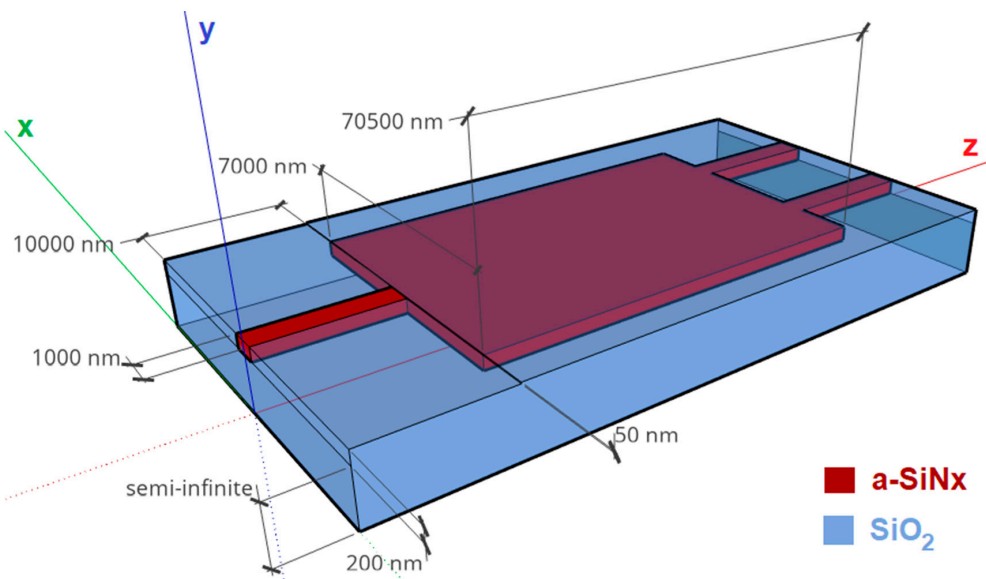

**Figure 1.** Diagram of simulated MMI structure and involved dimensions.

Moreover, by using an interference mechanism designated as symmetric interference, $1 \times N$ power splitters may be designed with quarter-length MMI sections. The mechanism relies on preventing the excitation of odd order modes within the multimode section of the structure. This is attained by placing the input waveguide, with a propagating symmetric field profile (e.g., Gaussian beam), at the width center of the MMI section, thus resulting in N linear combinations of the input profile located at:

$$L = \frac{p}{N}\left(\frac{3L_\pi}{a}\right),$$

(5)

where $p \geq 0, N \geq 1, p$, and $N$ are integers, have no common divisor and $p/N$ represents 1st, 2nd, ... , $nth$ single and N-fold images at location $L$; $a$ denotes the type of coupler (for a $1 \times N$ power splitter, $a = 4$).

## 3. Results

In this work, we performed the analysis of a symmetric interference MMI device tolerance to manufacturing deviations in CMOS photonics. The effective refractive index variations due to the limited resolution of lithographic mask process are known to adversely affect the splitting ratio from the optimum MMI device design. Hence, the width and length of our simulated MMI structure were designed as short as possible to assure the highest tolerance to manufacturing perturbations.

Previous studies [3,11] have been reported in the literature where an analytical approach based on a 2D paraxial approximation has been performed, considering bandwidth and tolerance analysis of InP MMI-based devices. Nevertheless, this method is applicable for any other suitable semiconductor, namely a-SiNx. Once the MMI section length is established, wavelength $\lambda_{opt}$ for optimal self-image generation may be expressed by combining Equations (4) and (5), resulting in Equation (6):

$$\lambda_{opt} = \frac{p}{N}\frac{4}{aL}\left(n_{eff}W^2\right).$$

(6)

Considering Equation (4), one is able to notice that the operating wavelength is inversely related to the beat length, thus longer wavelengths require shorter devices and vice-versa. Also, variations in the operating wavelength, width, effective refractive index, and length of the MMI coupler are related by:

$$\frac{\partial L}{L} = 2\frac{\partial W}{W} = \frac{\partial n_{eff}}{n_{eff}} = \left|\frac{\partial \lambda}{\lambda}\right|.$$

(7)

From Equation (7) it can be directly inferred that shorter devices will be more tolerant, thus higher losses and smaller fabrication tolerances are expected for a cross-coupler, when compared to a 3 dB MMI coupler. For well-separated output images, any design parameter variation leads first to an increase of the losses, so the optical bandwidth and the fabrication tolerances can be described in terms of excess losses, $\alpha$.

In Besse's work [3], excess losses are calculated as a function of the MMI length variation and the optical field amplitude of the single mode symmetrical, for both input and output waveguides, is approximated by Gaussian beams of waist $d_0$. Hence, MMI section length variation may be expressed by:

$$\partial L \leq Z(\alpha)d_0^2\frac{\pi n_{eff}}{\lambda},$$

(8)

where $Z(\alpha)$ is a function of the excess loss $\alpha$.

In previous article [3], the authors have expressed excess losses as a function of the structure length variation but considering the width perturbation as a tolerance-affecting factor, for this is the most critical geometrical parameter. Our work also assesses width variation but by evaluating the output power imbalance dependence to lithographic mask finite resolution. The simulated structure consists of input and output waveguides, and the MMI section affected by independent defects distributions along longitudinal edges.

Defects distribution standard deviation affects the width (*x*-axis) of all structure sections along the propagation length (*z*-axis) and are independent in each section of the structure (input and output waveguides and MMI section). It was assured that defects on the left and right longitudinal edges of each section were not correlated to one another, nor even within sections. The correlation length was 10 μm and 70.5 μm for input/output waveguides and the MMI section (full length of each section), respectively. This has been the utilized strategy to emulate defects distribution in real-world devices.

A silicon nitride waveguide buried in silicon dioxide is not a usual waveguide design. However, there are some reports in the literature which have implemented similar technology. Namely, for the manufacturing of such a device, the process could rely on a bottom thermal $SiO_2$ substrate, followed by Plasma Enhanced Chemical Vapor Deposition (PECVD) of the a-SiNx waveguide and an $SiO_2$ cladding which can be obtained by a PECVD process [13] or by plasma gas decomposition [14].

Figure 2a shows the impact on a 5:1 display ratio for better readability of lithographic mask defects on the refractive index of the MMI device, when considering 10 nm standard deviation for the random distribution of perturbations. This is a representation of the refractive index profile of a particular structure under previously referred constraints regarding defects distribution and at the y = 0.1 μm cut plane (half thickness).

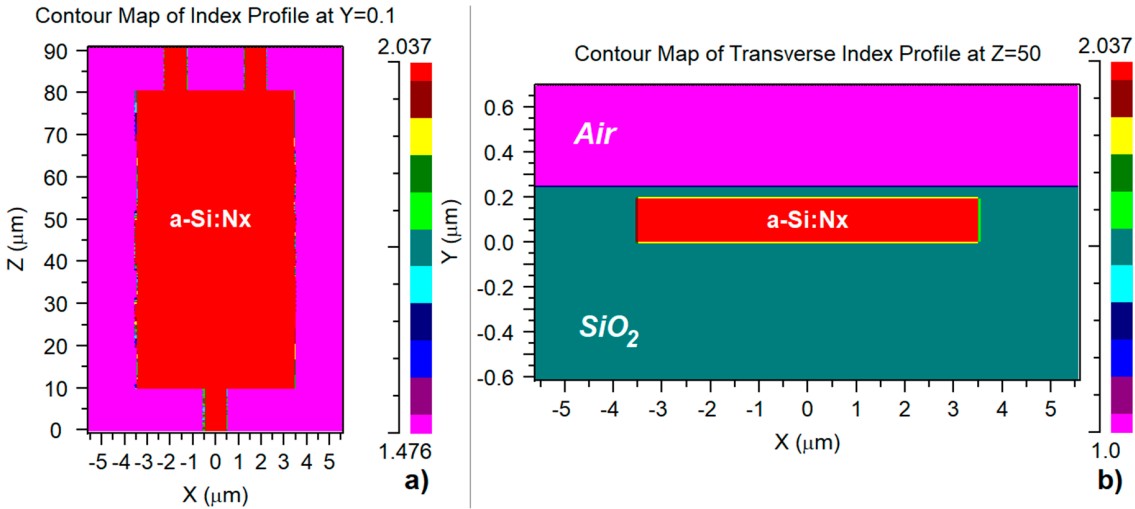

**Figure 2.** (**a**) Refractive index perturbations due to lithographic mask defects (10 nm standard deviation); (**b**) Representation of transversal refractive index profile at *z* = 50 μm (multimode interference section embedded in $SiO_2$).

Figure 2b depicts, on a 1:5 display ratio, previously mentioned structure refractive index profile on *z* = 50 μm cut plane, where one is able to point out features like its thickness at this location and both the $SiO_2$ substrate and same material 50 nm covering layer. Simulations were conducted on a device with the following characteristics and where the MMI section length was determined considering Equation (5) with corresponding parameters for a 3 dB power splitter device (*p* = 1, *N* = 2 and *a* = 4):

- 1 × 0.2 × 10 μm for input and output waveguides width, height and length, respectively;
- 7 × 0.2 × 70.5 μm for MMI section's width, height and length, respectively;
- MMI device is completely embedded in $SiO_2$ with a 50 nm cover; the superstrate is air.

Next, the imbalance between output waveguides is defined as the ratio expressed by Equation (9):

$$Imbalance = \frac{Out_2 - Out_1}{Out_2 + Out_1}. \tag{9}$$

Beam Propagation Method (BPM) simulations were conducted on a device with the characteristics presented in Figure 2, with the fundamental TM mode as the input field and at the operating wavelength

of 650 nm, while iterating standard deviation of defects distribution from 1 to 30 nm. Power imbalance between output waveguides was monitored at the 10 μm mark (the end of output waveguides), providing the results depicted in Figure 3. As can be observed and for this particular sample, measured power imbalance remains under the 10% boundary for manufacturing defects standard deviation up to 14 nm, increasing almost monotonically for higher order values. These standard deviation higher values may, in fact, compromise the MMI structure's main purpose, which is to generate two identical mode profiles in both detecting and reference arms for our biomedical-sensing device.

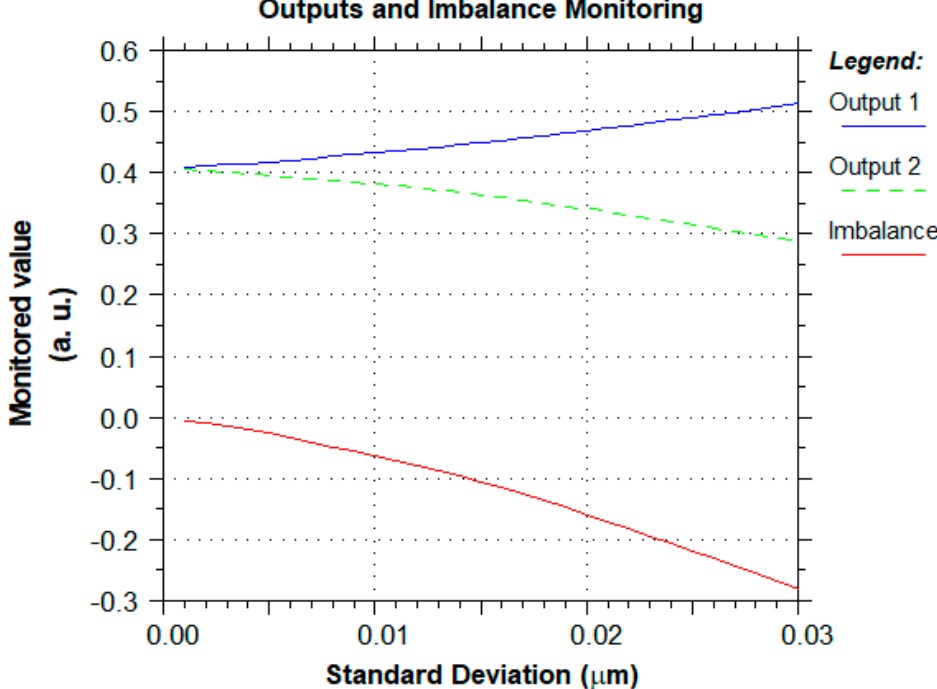

**Figure 3.** Resulting imbalance due to limited lithographic mask resolution.

Then, a slightly different approach was considered. Under a manufacturing perspective, we evaluated the influence of lithographic mask defects on a batch production of samples. To this end, we considered a manufacturing facility where 750 MMI devices were to be produced with limited lithographic mask resolution. All samples had random distributions of defects along their longitudinal edges, and all these perturbation distributions were independent to one another. The uncorrelated variables required to generate the random distributions were obtained through a discrete random function that is one of the features contained in Synopsys© simulation software tool [15].

To evaluate the tolerance to mask defects of the samples batch, BPM simulations were carried out while random perturbations' standard deviation was iterated from 1 to 15 nm. Figure 4 presents the imbalance verified along the last 5 μm of propagation in the output waveguides of all 750 samples. Verified field amplitude fluctuations are due to constructive/destructive interference caused by random refractive index perturbations, which result from the limited lithographic mask resolution along the longitudinal edges of the monitored output waveguide. A clear agglomeration of samples within the ±5% imbalance marks is distinguishable.

In order to quantify the amount of manufactured devices with imbalances between the output waveguides not above 5%, the imbalance of all 750 samples was computed considering an average of the last 5 μm of propagation in each sample. Then, statistical analysis of acquired data was performed and the results obtained show that, approximately, 93% of all produced samples are within this ratio range, as depicted in Figure 5.

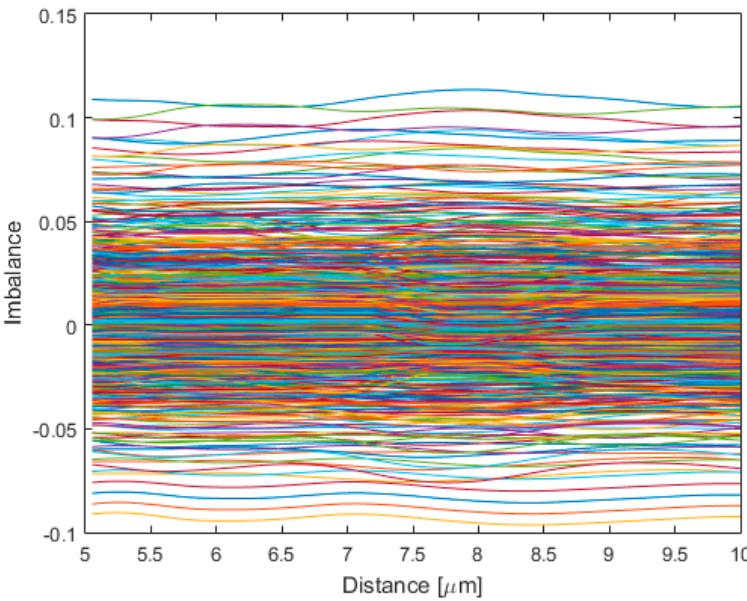

**Figure 4.** Imbalance along the last 5 μm of propagation in output waveguides for all 750 samples.

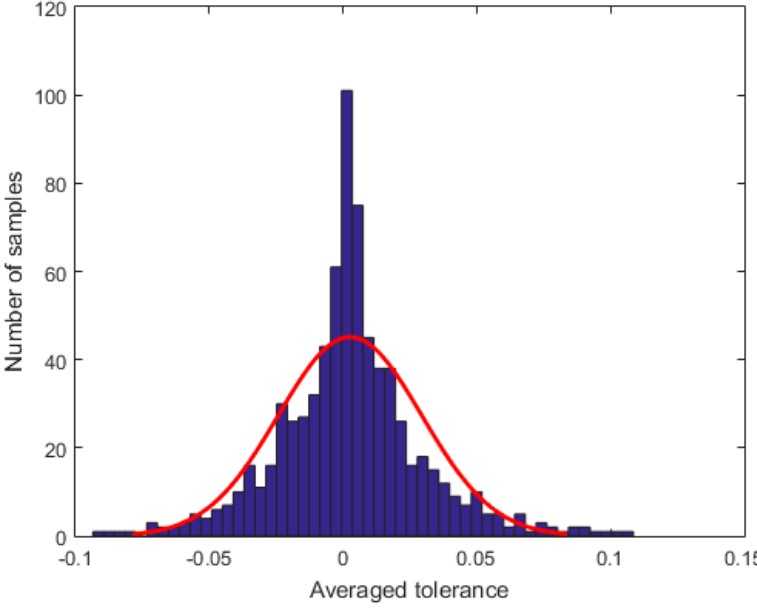

**Figure 5.** Imbalance histogram for the 750 manufactured devices.

## 4. Discussion

MMI devices have been used in many real life applications as splitters, combiners and couplers but usually on telecommunications operating wavelengths. In this work, by utilizing a-SiNx for the active semiconductor in our simulations, it has been shown that these structures may be designed to operate at other spectral ranges than telecommunications wavelengths. Moreover, these high index contrast structures enable the production of submicron cross section waveguides and small bending radii, which enables high-density integration. Device integration of many of such structures is achievable given design simplicity and involved minute footprints, thus complying with the lab-on-chip concept [9] when using these structures in PICs with integrated CMOS electronics. Nevertheless, lithographic mask defects may have a serious impact on these structures as demonstrated in this paper. In a production batch of 750 samples and considering the standard deviation of random perturbations introduced by

lithographic mask not higher than 15 nm, 7% of manufactured devices are expected to have power imbalance between the output waveguides above 5%.

Future work will consist in identifying subsequent actions that will contribute to mitigate the imbalance introduced by lithographic mask resolution, followed by simulations that will sustain the assumed concepts.

**Author Contributions:** J.C. has contributed with his simulation software expertise, A.F. has collaborated in the conceptualization and investigation, and together with M.V., provided the supervision and necessary resources to complete the work reported. Formal analysis, Investigation, Methodology, Writing—Original draft, P.L.

**Funding:** This research was supported by EU funds through the FEDER European Regional Development Fund and by Portuguese national funds by FCT—Fundação para a Ciência e a Tecnologia with projects PTDC/NAN-OPT/31311/2017, SFRH/BPD/102217/2014 and by IPL IDI&CA/2018/aSiPhoto.

**Conflicts of Interest:** The authors declare no conflict of interest.

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
