# Peer review of "Lithographic Mask Defects Analysis on an MMI 3 dB Splitter†"

_photonics, doi:10.3390/photonics6040118_

Round 1

Reviewer 1 Report

This paper present the results of a numerical study of the effects of imperfections introduced by manufacturing processes of MMI power splitters. Using realistic tolerances of mask defects, they claim that 93 % of the 3 dB power splitters have power imbalance less than 5 %. While this is an interesting result, there are several issues with the presentation that needs to be addressed. 

My main concern is the results presented in Figure 3 and 4. Both appear to be results of BPM simulations with different standard deviation of defect distribution. More details of the random variations of the mask defects should be given. Is the standard deviation given in the x-direction? What is the standard deviation and correlation length in the z-direction. 

The results in Figure 4 seems reasonable showing that the imbalance vary from device to device, with a mean imbalance of 0 % with a most devices falling between +/- 5 %. The results in Figure 3 on the other hand show that the imbalance increase with standard deviation. The standard deviation is a result of random variations, therefore the imbalance should also vary. So what do Figure 3 show?

Other issues that should be addresses includes:

line 73-78: Is the analysis equally valid for TE and TM modes? line 78-79: The effective index and effective width should be better defined.  Figure 1 should be redrawn and the figure caption should be more explanatory.  line 125: It is unclear what "previous article" refers to. line 140: What was the excitation field used for input? Line 141: How was the length 70.5 micrometer determined? Figure 2: I assume "a-Si:H" should be replaced by "a-SiNx" Figure 2: A silicon nitride waveguide core buried in silicon dioxide is an unusual waveguide design. This should be addresses. How could this be manufactured? line 151: How was the output power computed? With a distance of only 10 micrometer from the MM waveguide output, radiation modes could be an issue. line 172: A reference to the standard deviation (1-15 micron) of the mask defects should be given.  line 227: Ref 8 is incomplete. line 234: Ref 11 is incomplete. line 234: Ref 12 is NOT published in Optics Express.

Reviewer 2 Report

For the realization of multimode interference (MMI) 3dB power splitter in visible range with low losses and investigation of influence introduced by the limitation of manufacturing processed, detailed simulation and analyses are given in this manuscript.

As described in the manuscript, this is an extension of the AOP 2019. It is better to allow reviewers to access this paper before giving the final decision. Based on this version submitted, the major revised version is recommended.

My detailed comments are as follows:

From the abstract and introduction, it seems that the main purpose of this paper is characterizing the influence of defects introduced by manufacturing processes on the MMI power splitter. But the title of this paper highlights the visible range which is not the key point discussed in this paper. The author mentioned surface plasmon resonance (SPR) in abstract and listed it as a key word, but it did not appear once in the rest of this paper. In addition, the structure of MMI splitter proposed and simulated by the author does not contain any interface to generate SPR. The author mentioned in the abstract that the result is obtained by both FDTD and BPM method, but the simulation of FDTD are not reflected in the result and discussion sections. As author’s description in line 151, output energy 1 and 2, which determines the imbalance, are both obtained by applying BPM simulation at 650nm. In the discussion part line 186 and 187, author mentioned that it has been demonstrated that operation within the visible range is also attainable. However only the output energy difference between two output arms is shown and discussed in the discussion part. To support above conclusion, the device efficiency and comparison between input and output energy in visible band may be more convincing. The topic of section 2 is Materials and Methods (line 63), but there is no mention of material throughout this section. Fig.1 needs to add some comments; it could be more understandable if some dimension parameters can be shown in the figure. The axis labels in Fig. 3 is confused to readers, it is better not to use abbreviation at here. Fig. 4 and Fig. 5 need more specific axis labels. The equations in this article are not in proper format, they are all shown in low resolution.
